# Investigating the work–life experiences of nursing faculty in Canadian academic settings and the factors that influence their retention: protocol for a mixed-method study

Sheila A Boamah 

Nursing, McMaster University Faculty of Health Sciences, Hamilton, Ontario, Canada

**Correspondence to**
Dr Sheila A Boamah;
boamahs@mcmaster.ca

## ABSTRACT

**Introduction** While all research-oriented faculty face the pressures of academia, female faculty in fields including science, engineering, medicine and nursing, are especially susceptible to burnout. Nursing is unique in that it remains a predominantly female-dominated profession, which implies that there is a critical mass of females who are disproportionately affected and/or at higher risk of burnout. To date, little is known about the experiences of nursing faculty especially, new and early career researchers and the factors that influence their retention. This study aims to understand the work–life (the intersection of work with personal life) experiences of nursing faculty in Canadian academic settings and the factors that influence their retention.

**Methods and analysis** A mixed-method design will be used in this study. For the quantitative study, a sample of approximately 1500 new and early career nursing faculty across Canadian academic institutions will be surveyed. Eligible participants will be invited to complete a web-based structured questionnaire in both French and English language. Data will be evaluated using generalised linear regression model and structural equation modelling. Given the complexities of work–life issues in Canada, qualitative focus group interviews with about 20–25 participants will also be conducted. Emerging themes will be integrated with the survey findings and used to enrich the interpretation of the quantitative data.

**Ethics and dissemination** This study has received ethical approval from the Hamilton Integrated Research Ethics Board (#1477). Prior to obtaining informed consent, participants will be provided with information about study risks and benefits and strategies undertaken to ensure confidentiality and anonymity. The study findings will be disseminated to academics and non-academic stakeholders through national and international conference presentations and peer-reviewed open-access journals. A user-friendly report will be shared with professional nursing associations such as the Canadian Associations of Schools of Nursing, and through public electronic forums (e.g., Twitter). Evidence from this study will also be shared with stakeholders including senior academic leaders and health practitioners, government, and health service policy-makers, to raise the profile of discourses on the nursing workforce shortages; and women's work–life

## Strengths and limitations of this study

► This will be the first national study examining the work–life experiences of nursing faculty in Canada.
► By adopting a two-phase sequential explanatory mixed-method design, the proposed study will explore in-depth the dichotomy of work and non-work life and strategies to improve faculty retention; and more broadly, raise the profile on discourse on women's work–life balance in Canada and beyond.
► While the present study focuses on nursing faculty wellness and retention and is situated in Canada, the findings can inform policies and practices in other academic disciplines and countries facing similar challenges.
► The cross-sectional design will not conclusively support causation to the evidence of covariation in the study variables.
► The subjective nature of qualitative research has potential for recall bias; thus, critical attention should be paid to rigour and trustworthiness.

balance, a public policy issue often overlooked at the national level. Such discussion is especially pertinent in light of the disproportionate impact of COVID-19 on women, and female academics. The findings will be used to inform policy options for improving nursing faculty retention in Canada and globally.

## INTRODUCTION

The debate over the causes of the under-representation of women in academia seems to divide the potential causal factors into gender-based versus structural inequity. The presence and status of women faculty are 'a pressing national issue for reasons of social equity (or inequity) in access to, and rewards gained in' p.998, the workforce.[1] While all research-oriented faculty face the pressures of academia, female faculty are especially susceptible to burnout.[2–4] Female faculty in fields including science, engineering,

BMJ

medicine and nursing often struggle with balancing career and family, and gaining the respect of students and colleagues.[1 5 6] Nursing is unique in that it remains a predominantly female-dominated profession which implies that there is a critical mass of females who are disproportionately affected by this imbalance.[7 8] Additionally, most nurse academics come from a clinical background with little preparation for the complex faculty role and understanding of the distinct academic culture, language, expectations, values and behaviours.[9 10] Given the extensive history of nursing practice and education in Canada, there is still not a commensurate understanding of faculty retention.

Currently, nursing in Canada is plagued by recurrent faculty shortage. This shortage directly impacts the supply and demand of registered nurses in the clinical work environments and the ability of nurses to deliver high-quality patient care. In 2016, only 19.3% of faculty members employed by schools of nursing in Canada held a permanent position.[11] This shortage is fueled, in part, by a rapidly ageing workforce, an undersupply of doctorally prepared nurses to take their place, lack of qualified applicants and poor work environment.[12–15] About 40% of Canadian nursing faculty are over the age of 55% and 17.8% are eligible to retire.[11] The retirement projections of the ageing faculty limit the pool of doctorally prepared faculty as fewer nurses are applying for graduate programmes.[15–18] According to Canadian Associations of Schools of Nursing,[11] the number of graduate students currently enrolled in nursing schools is far less than what is required to meet the demands for advance practice in clinical settings and in academia, and this is contributing to exponential growth in unoccupied faculty positions across Canada. This is exacerbated by the fact that only 60% of nurses with doctorates, choose the academic role.[19] Across Canada, schools of nursing identified shortage of qualified applicants, non-competitive salaries, and lack of funds to create permanent positions as the three main factors limiting their ability to recruit new faculty.[19–21] This results in heavy workloads for employed faculty, their inability to train sufficient graduate students, lack of quality time for students in the programme, contributing to longer graduation periods, faculty burnout, work–life conflict (or lack of balance between work and life), dissatisfaction and further attrition.[17 22 23]

With the concurrent nursing faculty shortage and the impact of the COVID-19 pandemic on the nursing workforce, attracting and retaining quality faculty is extremely important to educational institutions as low faculty retention rate might create both monetary and academic consequences.[24] The predominant discourse on nursing faculty recruitment and retention centres on the realities of an ageing workforce, lucrative opportunities in the clinical and private sectors, heavy workloads, poor work environment, lack of organisational support[25–27] and unrealistic role expectations.[23 28 29] What is missing in this discourse is a nuanced understanding of the joint effect of multiple work–life issues on recruitment and retention of early

career researchers (ECRs)/academics, and comprehensive strategies to alleviate the shortage of faculty. Additionally, past studies on nursing faculty have mainly been in the US context with little focus in Canada.[26 30] There is limited understanding, at all administrative levels, regarding the impact of institutional work–life issues including excessive workloads, long working hours and lack of mentorship on nursing faculty members' satisfaction, and subsequent implications on their intentions to leave their job.[10]

Using Structuration theory[31] to explore the interplay between individuals and organisations, this study seeks to understand work–life (the intersection of work with personal life) experiences of nurse academics, including ECRs and the factors they identify as relevant to their success. Exploring the perspectives, concerns and experiences of nursing faculty, especially ECRs, is important in academia in terms of succession planning. Thus, the overall goal of this study is to investigate nursing faculty's work–life experiences in Canadian academic settings and the factors that influence their retention.

### Objectives

The primary objectives of this study are to: (1) assess the facilitators and barriers that influence new and early career faculty's successful transition to the academic role; (2) examine the personal and situational factors that jointly influence nursing faculty's satisfaction and intentions to remain in their jobs; (3) delineate the comparative experiences of female and male academics; (4) examine how institutional leadership impacts faculty's quality of work–life, and productivity; and (5) explore nursing faculty's understanding of work–life integration and balance.

### Theoretical perspectives

Guided by the overall goal, the proposed study is grounded in the *Structuration theory*,[31] and the relational interdependence of structure and agency (see figure 1). The use of

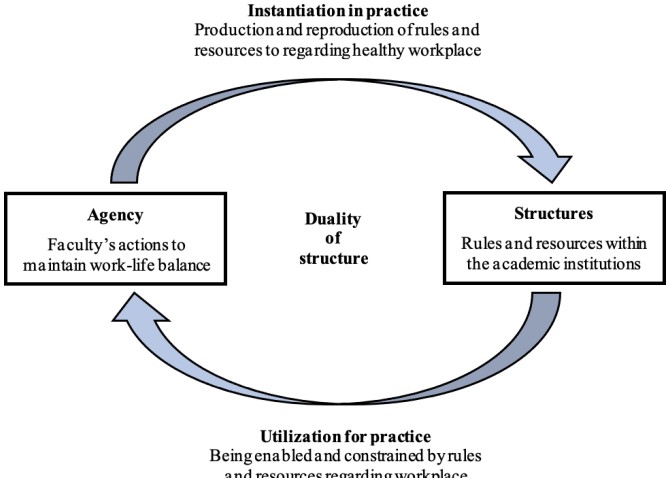

**Figure 1** Structuration model of the intersection of nursing academy and work–life balance (adapted from Giddens, 1984).

structuration as the theoretical lens in this research offers a multidimensional approach that takes into account the various underlying components of the 'structure' which is a set of rules and resources that enable and constrain decisions, choice, action and thought, and 'agency' referring to the ability of an individual to take action, with an appreciation for their interrelatedness.[32] Structuration theory takes a dualistic view of social worlds by considering both the members of the organisation (actors) and the organisation's structure, and the reality of the tensions within this dialectic and the dynamic and transformational processes inherent within organisations.[31] The actions that individuals in any organisation take to meet their work and non-work demands (eg, personal/family caregiving) at a particular time are shaped by the structural properties of that context (eg, resources, institutions).

Structuration theory is a powerful framework for studying the qualities of human relationships and a productive theoretical lens to apply to this study of work–life issues (or inter-role conflict) in academia particularly because it allows researchers to consider the different levels of influence that affect faculty's retention. Individuals' actions and interactions create or recreate the lower level structures (eg, at the family level), and therefore reinforce the higher level structures at the level of organisations.[33] In other words, individuals tend to create the kinds of family structures that fit the organisational expectations as they are created. Several researchers[33–35] have measured 'how' structures (eg, leadership) can influence employees' ability to be satisfied and productive at work. The literature indicates that leadership influences the alignment of individual behaviour in an organisation. Effective leadership has been shown to contribute to high morale in workplace which results in increased satisfaction and organisational commitment.[36–39] However,

it is unclear how institutional leaders may bring about behavioural changes in academia. This study seeks to address this gap and the evidence from this research will add to the theoretical basis for extending Giddens's structuration theory to incorporate leadership as a precursor of faculty satisfaction and retention.

The proposed study seeks to adapt and strengthen the use of this theory by exploring the role and effects of both structural/workplace and personal factors on individual's adjustment to the work environment, and the multiple dimensions of organisational and career satisfaction. By situating this study on faculty work–life within the structuration framework, evidence from this research on how the interplay of personal and situational factors culminate in either retention or attrition will have broader implications pertaining to the work–life debate among women in the Canadian workforce and beyond.

## METHODS AND ANALYSIS
### Design
This study will adopt a two-phase sequential explanatory mixed-method design—a cross-sectional survey followed by qualitative focus group interviews to answer the research objectives, given their combined strength in addressing the complexities of work–life issues in Canada.[40] The phases of the study including recruitment and data collection procedures are outlined in figure 2.

### Phase 1: quantitative methods and analysis
The first phase of the study will focus on the quantitative survey, which will help to address *Objectives 1, 2, 3,* and *4.* A comprehensive literature review on quality of work environment was conducted prior to the study which will be used to inform the survey design.

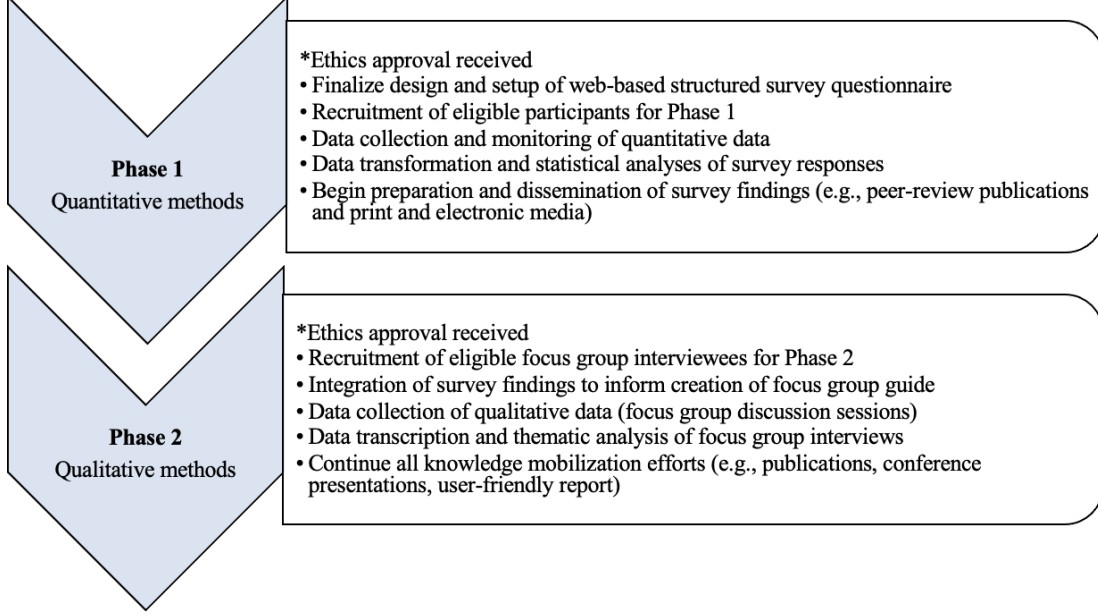

**Phase 1**
Quantitative methods

*Ethics approval received
- Finalize design and setup of web-based structured survey questionnaire
- Recruitment of eligible participants for Phase 1
- Data collection and monitoring of quantitative data
- Data transformation and statistical analyses of survey responses
- Begin preparation and dissemination of survey findings (e.g., peer-review publications and print and electronic media)

**Phase 2**
Qualitative methods

*Ethics approval received
- Recruitment of eligible focus group interviewees for Phase 2
- Integration of survey findings to inform creation of focus group guide
- Data collection of qualitative data (focus group discussion sessions)
- Data transcription and thematic analysis of focus group interviews
- Continue all knowledge mobilization efforts (e.g., publications, conference presentations, user-friendly report)

**Figure 2** Phases of study protocol.

## Sample

The population of interest is nursing faculty working in Canadian colleges and universities. Eligibility criteria include full-time or part-time instructional/teaching and/or research faculty (eg, lecturer, teaching track, and tenure-track professor), including ECRs within their first 5–7 years in academia. The 5–7 timeframe was chosen based on the tenure review process adopted by most Canadian universities.[41] Based on the estimated number of new and ECRs/academics in Canada,[19] a total of 1500 potentially eligible participants will be surveyed. While there is no defined formula for sample size estimation in structural equation modelling (SEM), a sample exceeding 200 subjects is recommended to maintain the accuracy of estimates, adequate power and to ensure representativeness.[42 43] As such, a targeted sample of 200 faculty members will be a sufficient sample size to obtain reliable parameter estimates among variables in the proposed theoretical framework for this study.[44]

## Recruitment and data collection

An email invitation will be sent to eligible faculty members using their publicly available e-addresses to complete a web-based structured questionnaire offered in both Canadian national languages—English and French. The questionnaire will be hosted securely online using the author's institutional site licensed Qualtrics survey software. The email invitation will include a description about the study, study purpose and inclusion criteria and a link to the online survey. The survey will contain a letter explaining the study risks and benefits as well as description of strategies undertaken to ensure confidentiality and anonymity. No personal identifiers will be collected and/or linked to individual respondents as participants will be assigned with a random PIN number to complete the survey anonymously. Participants can terminate the survey at any time as they desired prior to submitting their response without penalty. Return of a completed survey will indicate informed consent to participate. The tailored design method (TDM) for Internet surveys[45] will be used as a technique to improve survey response rates. The TDM involves a set of procedures intended to create respondent trust and perceptions of increased rewards with the overall goal of reducing survey error. Non-respondents will receive a reminder notice 3 weeks following the date of the initial invite, and a subsequent remainder message 4 weeks thereafter to improve participation and ensure an adequate sample size and representative sample. Other recruitment strategies and incentives will be considered including a chance for participants to enter a draw to win a prize (eg, gift card). At the end of the survey once participants have submitted their responses, they will receive a thank you note and an option to either exit the Qualtrics platform (eg, close the browser), or click on a separate link to

be redirected to a GoogleDoc form where they will provide informed consent if they choose to partake in the qualitative phase of the study (focus group) at a later date.

## Measures

Among other broad topics, the study survey will cover workload issues, mentorship, leadership practices, organisational support and job satisfaction. Examples of standardised questionnaires to be used in this study include the well-validated Multifactor Leadership Questionnaire,[46 47] Job Satisfaction Survey[48] and Work Interference with Personal Life (WIPL).[49–51] Sample items from the WIPL include: 'My job makes it difficult to maintain the kind of personal life I would like' and 'I often neglect my personal needs because of the demands of my work.' Issues of reliability and validity as well as survey length to minimise respondent burden will be considered in selecting the final survey items. Demographic data including age, gender, ethnicity, highest education, academic rank and tenure status will be collected. Anticipated time to completion the survey is approximately 15–20 minutes.

## Data analysis and interpretation

Descriptive statistics will be used to profile the characteristics of nursing faculty, and to analyse bivariate relationships using SPSS (v.26) software.[52] Rigorous methods will be used to verify the accuracy and quality of the measurement tools by performing confirmatory factor analysis and Cronbach's alpha reliabilities. Given the complexity of the research questions, the target is to employ SEM technique with maximum likelihood estimation to test the fit between the data and the hypothesised model in Mplus software (v.7.3).[44 53] SEM is a powerful multivariate technique which allows us to examine multiple relationships among variables in a single model.[54]

## Phase 2: qualitative methods and analysis

A major critique of quantitative surveys, including the web-based self-report survey, on work–life issues is that they do not provide in-depth accounts of the work experiences of faculty.[40] Therefore, in second phase of this study, qualitative descriptive research (focus group interviews) will be undertaken, to complement and enrich the quantitative findings by probing in greater detail the circumstances leading to potential turnover. This component of the study will address *Objective 5*.

## Recruitment and data collection

Focus group discussions have been characterised as a microcosm of 'the thinking society', capable of revealing the processes whereby social norms are collectively shaped through debate and argument.[55 56] As such, the proposed study will make use of focus group discussion. The semistructured focus group sessions will provide a forum for faculty members to

critically discuss a range of issues including work–life balance, workload demands, supports and resources. The sampling strategy for the focus group will consist of a purposive sample of approximately 20–25 survey respondents who volunteer to participate in the qualitative phase of the study. This is sufficient sample size for the type of design.[40] There will be five focus groups, consisting of a mix of all gender including male and female nurse academics. Each focus group discussion will have five members (n=25) and will be about 60 minutes in duration. The number of focus group attendees in this study is informed by the requirements of thematic/theoretical saturation, which is the point at which themes become repetitive.[57] Faculty members will be interviewed[58] virtually/online via Zoom or any other communication medium of their choice to enhance the feasibility of this phase given the national scope of the study. There will also be an option for in-person interview as face-to-face interviews offer the advantages of rapport and visual cues.[58] As a token of appreciation, focus group participants will receive a $25 e-gift card.

### Focus group interview guide

An interview guide will be developed stemming from the phase 1 of the study. Sample focus group questions include: 'Tell us about your first few months/years as a new professor' and 'What do you believe has had the most impact on your transition to the academic role?' After obtaining informed consent from participants, the focus group discussion will be audio recorded using Audacity and transcribed verbatim, and the research team will perform a textual analysis of emerging themes and conceptual categories, while also looking for patterns and relationships.

### Data analysis and interpretation

Initially, data analysis will focus on understanding the information, then identifying codes and categories. Thematic analysis will be conducted using open-coding techniques.[59] Codes will be applied to statements that share commonalities. Topics of discussion in response to questions, areas of agreement and disagreement, and characteristics of the discussion will be examined.[60] In addition to themes that emerge from the data, other themes (eg, demands of teaching, and lack of clarity about tenure requirements) not included in the preliminary framework will be included in analysis inductively and the framework will be modified accordingly. The qualitative data analysis software, N-Vivo (v.12),[61] will be used extensively for data management and coding of text. Due attention will be given to rigour in this study. To maintain a high level of quality, methodological (interviews and survey) and theory triangulation will be used, including keeping audit trails, and engaging in biweekly debriefing sessions. Data from the observations and interviews will be anonymised, stored securely at the institutional research data repository throughout the data life (5 years).

### Patient and public involvement

Patients were not involved in the design, or conduct, or reporting, or dissemination plans of this research.

## ETHICS AND DISSEMINATION

This study has been approved by the Hamilton Integrated Research Ethics Board (HiREB) (#1477). To ensure the privacy, dignity, integrity and anonymity of the participants, only group data will be reported. Prior to consenting to the study, participants will be provided with information about study risks and benefits and strategies undertaken to ensure confidentiality and anonymity. Participants will have the option of terminating the survey at any time as they desired prior to submitting their response and/or decline to partake in the focus group.

To ensure the findings have wide accessibility and impact, the knowledge mobilisation (KM) plan for this study involves diverse strategies (eg, stakeholder engagements, publications, conferences). Target audiences for this project are policy-makers and influencers (eg, government officials, institutional leaders/administrators, practitioners, academics, health and labour unions). KM will be ongoing throughout the project to provide these audiences with timely information about the project and create meaningful exchange of ideas among these groups. Findings from this study will be presented at academic and non-academic conferences, through open access peer-review publications, and media release to reach the widest audiences, including interested laypeople and the general public. Academic and international audiences will be reached mainly through journal publications and conference presentations and a user-friendly report will be shared with nursing associations and health agencies. Further, social media (e.g., Twitter) will be used for connecting and networking these audiences.

## DISCUSSION

Although there are emerging studies on nursing faculty work–life,[2 18 26] the perspectives and experiences of new and ECRs are largely missing from the current debates on the issue. This perspective is vital because early career faculty are the next generation of leaders in academia, and they can provide valuable new insights into the successes and challenges of retention in the long term. Considering the views of new faculty brings to the fore the necessity of thinking in terms of succession planning. Although the nursing faculty shortage is a multifaceted problem, a few existing studies of the crises are descriptive in nature and most studies are limited to single sites.

The present study is innovative in a number of levels: first, it aims to combine quantitative and qualitative methodologies in novel ways to gain a nuanced understanding of faculty members' work–life experiences and how

academic leadership influences new faculty recruitment and retention. Second, the focus on academic leadership is unique in that less is known about how leadership of nursing faculties influence new nurse academics' quality of work–life, satisfaction and subsequent retention.[18] Third, this research aims to examine the dynamic interplay of leadership, work environment and satisfaction by jointly focusing on: (1) individual-level characteristics including personal and psychological traits; (2) structural characteristics including institutional support and culture, and (3) psychosocial attributes (eg, coping, support) to provide sophisticated and comprehensive view of the transitional experiences of new faculty and resources that will aid in their success. The existing research on nursing faculty experiences is in the US context and very little on the Canadian context. Furthermore, there is paucity of research on the ECRs' experiences and how that culminates in retention.

The proposed study will make substantive contributions to both research and practice on nursing faculty recruitment and retention, which can serve as a baseline to address the national and global nursing faculty shortage. The findings will expand knowledge on how leadership impact retention decision making in the academic context, as well as in the organisational literature. The outcomes of this study will be valuable for nursing faculty planning and decision making in the medium to long term; and will advance work–life research by using evidence-based planning to explore informed policy-making, at a broader, conceptual level and in the specific case of recruiting and retention as a proxy for local and national policies. Furthermore, findings from this study can inform policy development, implementation and evaluation process.

While this study focuses on faculty recruitment and retention and is situated in Canada, the goal is to develop an informed policy-making framework that is applicable in the international contexts, especially countries facing similar challenges. More fundamentally, the proposed research will raise the profile of discourses on women's work–life balance, a public policy issue often overlooked at the national level. Such discussion is especially pertinent in light of the disproportionate impact of COVID-19 on women, and female academics. Specifically, this study will highlight the critical role of informed/evidence-based planning plays in society, and its importance for stabilising and enhancing recruitment and retention of new faculty. The findings will make an original contribution to work–life studies by shedding light on important challenges, considerations and strategies for addressing the current nursing faculty shortage in Canada. Overall, the proposed study will contribute to a gap in the literature and enhance the current evidence on nursing education and faculty well-being in Canada.

This study has certain limitations to acknowledge in relation to the design, analysis and generalisability. First, the cross-sectional design precludes the ability to make statements of cause and effect to the evidence of covariation in the study variables and the *a*-priori theoretical associations. Additionally, the use of self-reported measures has potential for response bias.[62] Common method variance may also be of concern when the same individual completes all measures, and so, multiple-item and well-designed scales will be used to decrease the likelihood of this.[62] Another potential limitation is that the data will be collected from Canadian nursing faculty, which may limit generalisability to faculty members in all disciplines and/or countries. Additionally, there are numerous other factors not included in this study that could account for nursing faculty retention. For instance, personal dispositional variables, such as resilience or coping self-efficacy, may play an important role in addition to the study variables. Lastly, the subjective nature of qualitative research has potential for recall bias; thus, critical attention should be paid to rigour and trustworthiness. Further research is required to address these limitations.

**Acknowledgements** The author wishes to thank Dr. Arku for reviewing the initial manuscript draft.

**Contributors** SAB conceptualised the design of the study, drafted and revised the manuscript.

**Funding** This work was supported by the Social Science and Humanities Research Council (SSHRC) of Canada, grant number [430-2020-01042].

**Competing interests** None declared.

**Patient and public involvement** Patients were not involved in the design, or conduct, or reporting, or dissemination plans of this research.

**Patient consent for publication** Not applicable.

**Provenance and peer review** Not commissioned; externally peer reviewed.

**ORCID iD**
Sheila A Boamah http://orcid.org/0000-0001-6459-4416

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
