## [Reviewer comments · BMJ Open]

ARTICLE DETAILS

TITLE (PROVISIONAL)	Investigating the work-life experiences of nursing faculty in Canadian academic settings and the factors that influence their retention: protocol for a mixed- methods study
AUTHORS	Boamah, Sheila A.

VERSION 1 – REVIEW

REVIEWER	Purdy, Nancy Ryerson University, Nursing
REVIEW RETURNED	04-Oct-2021

GENERAL COMMENTS	The goal of the proposed study is to investigate nursing faculty's worklife experiences in Canadian academic settings and the factors that influence their retention. As a first study to examine faculty experiences across Canada, the study is needed and timely given the pressures to address the global nursing shortage with increasing nursing student enrollment targets, the expansion of institutions offering undergraduate nursing degrees, the aging nursing faculty workforce and the resulting need to recruit and retain nursing faculty. Worklife experiences will be described in a comprehensive manner including an exploration of personal as well as organizational facilitators and barriers, work and non-work strategies, a gender comparison and a description of faculty member's transition to academia. A mixed-method study design is appropriate to achieve the study objectives. Gidden's structuration theory will be used to inform the study as it acknowledges the interaction of personal and structural factors. This theory, as described in the proposal, is less explicit in identifying specific worklife factors that lead to faculty satisfaction and retention than other frameworks such the Tourangeau et al. (2014) conceptual model of Factors Influencing Nurse Faculty Intention to Remain Employed that was used to examine nursing faculty worklife in Ontario. Either a more explicit model (e.g. revised Figure 1) or the Tourangeau model would help to inform the hypothesized study model and ensure better alignment of the study objectives, selected variables, and data collection details than is currently described. The sample will include nursing faculty hired within the last seven years. It would be useful to confirm if the sample will include only part time and full time faculty so that the work expectations are more homogenous e.g. demands of tenure in a university setting or teaching and administrative demands for college settings. A sample size of 200 is justified but may be challenging to obtain given current work demands. While using the Dillman method is useful, other recruitment strategies may need to be employed to achieve the required sample size. It would be useful to include the names of all 7 standardized instruments to better
--

	understand the variables of interest. Alternatively, including the hypothesized study model will also add clarity for the quantitative phase of the study. The subsequent qualitative phase of the study will be an important adjunct to the quantitative results. Involving 20-25 participants from phase 1 in a set of focus groups is a useful plan but perhaps using an online method for all focus groups could enhance the feasibility of this phase given the national scope of the study. The data analysis and knowledge management strategies are appropriate. Overall, this proposed study has the potential to influence both personal and organizational strategies to reduce nursing faculty burnout, improve working experiences to retain faculty and to inform broader work strategies for other female academics. Minor edits suggested:  -clarification, you refer to 'we' in the proposal but only one author is listed -page i, line 9-10 – it is not clear what is meant by 'this imbalance' -page i, line 36 – suggest a reference for the statement about the impact of COVID on nursing faculty -page 3, line 40 – the reference is from 2014, is there a newer source from CASN? -page 3 – line 54 – you state that you will not include personal identifiers in the online survey but you will need to ask for participants to volunteer for the focus groups so you may need to revise this statement -page 4 – line 52 – volunteer (not volunteers) -page 5, line 25 – emerge (not emerges) -page 6, lines 8-14 – these research aims do not seem to align with the study objectives stated earlier or in the study instruments section. Suggest better alignment throughout (as noted earlier). -page 6, line 25-28 – unsure what you mean by the last sentence in this paragraph -page 17, Figure 2 – suggest removing 'e.g. coding' as you are describing the quantitative phase
--	--

REVIEWER	McPherson, S University of Illinois at Chicago
REVIEW RETURNED	14-Oct-2021

GENERAL COMMENTS	This is an excellent and very timely topic. Understanding the academic process is difficult and recruiting and retaining quality faculty are often a challenge. The results of this study would allow for a better understanding of women in academic nursing. This is important as we work to recruit more faculty. Nurses have many choices today and this study will highlight areas for improvement in the academic setting.
---

VERSION 1 – AUTHOR RESPONSE

Reviewer 1

Comments to the Author:

The goal of the proposed study is to investigate nursing faculty's worklife experiences in Canadian academic settings and the factors that influence their retention. As a first study to examine faculty experiences across Canada, the study is needed and timely given the pressures to address the global nursing shortage with increasing nursing student enrollment targets, the expansion of institutions offering undergraduate nursing degrees, the aging nursing faculty workforce and the resulting need to recruit and retain nursing faculty. Worklife experiences will be described in a comprehensive manner including an exploration of personal as well as organizational facilitators and barriers, work and non-work strategies, a gender comparison and a description of faculty member's transition to academia.

Response 1

Thank you very for the general comment

Comment 2

A mixed-method study design is appropriate to achieve the study objectives. Gidden's structuration theory will be used to inform the study as it acknowledges the interaction of personal and structural factors. This theory, as described in the proposal, is less explicit in identifying specific worklife factors that lead to faculty satisfaction and retention than other frameworks such the Tourangeau et al. (2014) conceptual model of Factors Influencing Nurse Faculty Intention to Remain Employed that was used to examine nursing faculty worklife in Ontario. Either a more explicit model (e.g. revised Figure 1) or the Tourangeau model would help to inform the hypothesized study model and ensure better alignment of the study objectives, selected variables, and data collection details than is currently described.

Response 2

Thank you for your suggestion. While Tourangeau et al.'s (2014) conceptual framework measured similar variables proposed in this study, it was poorly descriptive and has never been re-tested. For a study of this caliber and magnitude, it is important that I draw from a well-established and tested theory. As such, the Gidden's structuration theory (ST) was chosen intentionally for the purpose of this study as it has been validated in various organizational studies and shown to be effective in providing deeper understanding of the interplay between individuals and organizations, and how this relationship develops and provides opportunity for organizational analysis and change. This is a useful theoretical lens to apply to the study of worklife issues in academia particularly because it allows me to consider the different levels of influence that affect individuals' worklife experience. It is worth noting that the ST received great review from the funding agency during the peerreview process. Moreover, since several key variables are being considered in the study, I will propose and empirically test different hypothesized models/pathways.

Comment 3

The sample will include nursing faculty hired within the last seven years. It would be useful to confirm if the sample will include only part time and full time faculty so that the work expectations are more

homogenous e.g. demands of tenure in a university setting or teaching and administrative demands for college settings.

Response 3

Thank you for your comment. The manuscript has been revised accordingly. Please see page 3, paragraph 3 (see Sample).

Comment 4

A sample size of 200 is justified but may be challenging to obtain given current work demands. While using the Dillman method is useful, other recruitment strategies may need to be employed to achieve the required sample size.

Response 4

Thank you. The manuscript has been revised. Please see page 4, paragraph 1.

Comment 5

It would be useful to include the names of all 7 standardized instruments to better understand the variables of interest. Alternatively, including the hypothesized study model will also add clarity for the quantitative phase of the study.

Response 5

Thank you for your suggestions. I have provided examples of the instruments that I have copyright/permission to use at this time and mentioned the constructs that I intend to measure. Since this is the study protocol, I want to provide a high-level synthesis and so, there will be more than a singularly hypothesized study model to test once the study is completed.

Comment 6

The subsequent qualitative phase of the study will be an important adjunct to the quantitative results. Involving 20-25 participants from phase 1 in a set of focus groups is a useful plan but perhaps using an online method for all focus groups could enhance the feasibility of this phase given the national scope of the study.

Response 6

Thank you for your comments. The manuscript has been revised accordingly. Please see page 5, paragraph 2 (see Recruitment and data collection).

Comment 7

The data analysis and knowledge management strategies are appropriate.

Response 7

Thank you.

Comment 8

Overall, this proposed study has the potential to influence both personal and organizational strategies to reduce nursing faculty burnout, improve working experiences to retain faculty and to inform broader work strategies for other female academics.

Response 8

Thank you very much.

Comments 1-10

Minor edits suggested:

-clarification, you refer to 'we' in the proposal but only one author is listed - ✓

-page i, line 9-10 – it is not clear what is meant by 'this imbalance' - ✓

-page i, line 36 – suggest a reference for the statement about the impact of COVID on nursing faculty - ✓

-page 3, line 40 – the reference is from 2014, is there a newer source from CASN? - ✓

-page 3 – line 54 – you state that you will not include personal identifiers in the online survey but you will

need to ask for participants to volunteer for the focus groups so you may need to revise this statement - ✓

-page 4 – line 52 – volunteer (not volunteers) - ✓

-page 5, line 25 – emerge (not emerges) - ✓

-page 6, lines 8-14 – these research aims do not seem to align with the study objectives stated earlier or in

the study instruments section. Suggest better alignment throughout (as noted earlier). - ✓

-page 6, line 25-28 – unsure what you mean by the last sentence in this paragraph - ✓

-page 17, Figure 2 – suggest removing 'e.g. coding' as you are describing the quantitative phase - ✓

Response 8

Thank you very much for your comments. The manuscript has been revised accordingly

Reviewer 2

Comment 1

This is an excellent and very timely topic. Understanding the academic process is difficult and recruiting

and retaining quality faculty are often a challenge. The results of this study would allow for a better understanding of women in academic nursing. This is important as we work to recruit more faculty.

Nurses

have many choices today and this study will highlight areas for improvement in the academic setting.

Response 7

Thank you very much for your comments. Much appreciated

VERSION 2 – REVIEW

REVIEWER	Purdy, Nancy Ryerson University, Nursing
REVIEW RETURNED	14-Dec-2021

GENERAL COMMENTS	The author has provided a considered and thoughtful response to my questions and made suggested edits where necessary. There are only a few minor edits that are needed for the revised version of this manuscript. -page i, line 50 – Web-based... - web does not need to be capitalized-page 4, line 15 – ‘return’ should be plural OR you could delete this last phrase “and to maximize...” as it is redundant Return’-page 4, line 24 – revise ‘exist’ to ‘exit’-page 6, line 46 – sentence incomplete, “Further, social media (e.g., Twitter) will be.....
--